# Development of Azo Dye Immobilized Poly (Glycidyl Methacrylate-Co-Methyl Methacrylate) Polymers Composites as Novel Adsorbents for Water Treatment Applications: Methylene Blue-Polymers Composites

**DOI:** 10.3390/polym14214672

**Published:** 2022-11-02

**Authors:** Mohamed R. El-Aassar, Tamer M. Tamer, Mohamed Y. El-Sayed, Ahmed M. Omer, Ibrahim O. Althobaiti, Mohamed E. Youssef, Rawan F. Alolaimi, Emam F. El-Agammy, Manar S. Alruwaili, Mohamed S. Mohy-Eldin

**Affiliations:** 1Chemistry Department, College of Science, Jouf University, Sakaka 2014, Saudi Arabia; 2Polymer Materials Research Department, Advanced Technology and New Materials Research Institute (ATNMRI), City of Scientific Research and Technological Applications (SRTA-City), New Borg El-Arab City 21934, Alexandria, Egypt; 3Department of Chemistry, College of Science and Arts, Jouf University, Saudi Arabia; 4Computer Based Engineering Applications, Informatic Research Insitute (IRI), City of Scientific Research and Technological Applications (SRTA-City), New Boarg El-Arab City 21934, Alexandria, Egypt; 5Physics Department, College of Science, Jouf University, Sakaka 2014, Saudi Arabia

**Keywords:** water treatment, immobilization, MB, sulphonated, poly glycidyl methacrylate, composites, metal ions removal, dichromate ions, permanganate ions

## Abstract

Methylene blue azo dye (MB) immobilized onto Poly (glycidyl methacrylate-Co-methyl methacrylate), (PGMA-co-PMMA), and sulphonated Poly (glycidyl methacrylate-Co-methyl methacrylate), (SPGMA-co-PMMA), polymers composites have been developed as novel adsorbents for water treatment applications. The effect of copolymer composition and sulphonation on the MB content has been studied. Maximum MB content was correlated to the Polyglycidyl methacrylate content for both native and sulphonated copolymers. Furthermore, the effect of the adsorption conditions on the MB content was studied. Sulfonated Poly (glycidyl methacrylate; SPGMA) was the most efficient formed composite with the highest MB content. The developed composites’ chemical structure and morphology were characterized using characterization tools such as particle size, FTIR, TGA, and SEM analyses. The developed MB-SPGMA composite adsorbent (27 mg/g), for the first time, was tested for the removal of Cr (VI) ions and Mn (VII) metal ions from dichromate and permanganate contaminated waters under mild adsorption conditions, opening a new field of multiuse of the same adsorbent in the removal of more than one contaminants.

## 1. Introduction

Many industries, such as food, paper, carpet, and mainly textile, use different dyes types: ionic, and non-ionic. As a result, wastewater resulting from such industries is polluted with dyes. Ten thousand tons are consumed annually worldwide, and at least a thousand tons are released in the wastewater from these industries [1,2]. 

Textile dyes are considered the most threatening source among other dyes [3,4]. The negative impact of releasing colors into the water system ranges from direct ones on aquatic life, and indirect ones on humankind’s life. Methylene blue (MB) is a cationic dye commonly used worldwide as a coloring material [5]. Accordingly, corrective action to remove dyes from wastewater is needed. Different techniques have been used to remove dyes from wastewater, including physical, chemical, and even biological [6]. Among these techniques, physical and chemical ones have been shown to be the most compromising techniques from many points of view [7,8,9,10]. Sustainability and cost-effective characteristics have driven the research for exploring alternative adsorbents for the last decades. Many publications have been published on the removal of different dyes [11,12,13]. The physiochemical technique attracts much attention among different techniques [14,15,16,17,18,19,20,21]. The physical adsorption of the dyes or chemicals bound over the surface of the adsorbent surface through electron exchange is the responsible interaction in the removal process [22]. Different factors affect the adsorption efficiency. Some of them are related to the structure of the adsorbent, and others are related to the operational conditions [23]. 

The adsorption technique is generally the most favored in the dye removal process, based on many criteria [7,8,9,10,11,12,13,14,15,16,17,18,19,20,21,22,23,24]. In this context, activated carbon has shown a wide application [25]. Khan et al. investigated a novel, green, and economical dual-functionalized pullulan/kaolin hydrogel nanocomposite (*f*-PKHN), which was fabricated and subsequently applied for the liquid-phase decontamination of paracetamol (PCT), a pharmaceutical pollutant [26]. However, different polymer-based adsorbents, such as grafted cotton fabrics [27], carboxylated alginate beads [28], and Phosphoric Acid Doped Pyrazole-g-Polyglycidyl Methacrylate [29], have been investigated in the removal of methylene blue dye from contaminated waste water. 

Equally, environmental pollution with heavy metals is pervasion worldwide with advances in industry. Many heavy metals, such as nickel, copper, cadmium, manganese, and chromium, are the most familiar toxic heavy metals used and the prevalence of environmental contaminants [30,31]. Low concentrations of those metals are essential as enzymes’ co-factors, while high concentrations cause high toxicity to the living cells by inhibiting metabolism.

Potassium permanganate is commonly used in multidiscipline processes as a potent oxidizing agent for the oxidative treatment of many organic and inorganic compounds in soil and water solutions [32,33]. To our knowledge, few publications have addressed the removal of permanganate ions. Adsorption is considered to be a cheap and efficient method for the removal of Mn (VII) from wastewater, using different adsorbents such as activated orange peel powder [34], activated carbon [35,36], *Prosopis cineraria* leaf powder [37], and millet husk [38].

Chromium can exist mainly as Cr (VI) or Cr (III) in the natural environment. Cr (III) species are less soluble and more stable compared to Cr (IV) species, which are highly soluble and mobile in aqueous solutions [39]. Chromium (VI) also has higher mobility than chromium (III); therefore, it has a more significant potential to contaminate groundwater. The high risk of chromium (VI) is associated with its high reactivity and potential carcinogenic properties [40]. Acute exposure to Cr (VI) causes nausea, diarrhea, liver, and kidney damage, dermatitis, internal hemorrhage, and respiratory problems [5]. Inhalation may cause acute toxicity, irritation, and ulceration of the nasal septum and respiratory sensitization (asthma) [41]. Ingestion may affect kidney and liver functions. Skin contact may result in systemic poisoning damage, severe burns, and interference with the healing of cuts or scrapes. If not treated promptly, this may lead to ulceration and severe chronic allergic contact dermatitis. Eye exposure may cause permanent damage. Adsorption is also considered to be a cheap and efficient method for the removal of Cr (VI) from wastewater using different adsorbents such as charcoal [42], activated carbon from various sources [43,44,45], polyaniline and its composites [46], and Chitosan [47].

Recently, Nicoleta Mirela Marin [48] developed natural and acrylic polymers with diazo acid Blue 11 (AB 113) as a chelating reagent to selectively remove metal ions; Zn^2+^, Mn^2+^, and Cr^3+^ from acid polluted wastewater. 

The present study’s novelty is the first emphasis on using one adsorbent to remove multi-contaminants from wastewater. This goal has been achieved as follows. Native PGMA and SPGMA-based copolymers have been immobilized with azo dye through the removal of methylene blue (MB) molecules as the first contaminates to develop novel composites adsorbents having an affinity for removal of Cr (VI) and Mn (VII) metal anions from dichromate and permanganate contaminated waters as second contaminates. The study focuses on the MB adsorption conditions affecting the formation of MB-SPGMA polymer composites. The copolymer formation was verified by FTIR and TGA analyses. Moreover, the effect of the copolymer composition on its particle size and morphology was investigated. Finally, the MB-SPGMA composite novel adsorbent was examined by removing Cr (VI) and Mn (VII) ions from dichromate and permanganate-contaminated waters.

## 2. Materials and Methods

### 2.1. Materials

Methyl Methacrylate (MMA) and Glycidyl methacrylate (GMA) were purchased from ACROS (USA), and Potassium persulfate and sodium bisulfite were obtained from Sigma Chem. Co. (St. Louis, MO, USA), Ethanol absolute perused from Adwic, Egypt, and finally, MB from Aldrich, Germany was perused. Potassium dichromate (K_2_Cr_2_O_7_), minimum assay 99%, was supplied by Sigma Aldrich, Germany. Potassium permanganate (KMnO_4_), a minimum assay of 99%, was supplied by Sigma Aldrich, Darmstadt, Germany.

### 2.2. Polymerization Process

Glycidyl methacrylate, methyl methacrylate, and their mixture were polymerized under fixed conditions to prepare polyglycildyl methacrylate, polymethyl methacrylate, and poly (glycidyl methacrylate-*co*-methyl methacrylate) with varied composition [49]. The monomer (W_0_) was dissolved in 0.05 M KPS solution in ethanol/water (1:1) and mixed well. The mixture was kept in a water bath at temperature of 60 °C for 3 h to polymerize and then left overnight at RT. white ppt was formed. The formed polymer was filtered and successively washed with ethanol/water solution to remove unreacted monomers and initiator. The polymers were dried overnight at temperature of 80 °C (W_1_). The polymer yield (%) was calculated according to the following equation:Polymer Yield (%) = (W_1_/W_0_) × 100(1)

### 2.3. Sulphonation Process

0.5 g of the polymer was reacted with 20 mL of 3% sodium sulfite (S.S.) solution (ethanol/water) at temperature of room for one hour. The sulphonated polymer was washed with ethanol/water to remove the unreacted S.S. The polymer was then dried at temperature of 60 °C overnight [50].

### 2.4. Particle Size Analysis

Particle size distribution was performed using Submicron Particle Size Analyzer (Beckman Coulter-Pasadena, Pasadena, CA, USA) through a dispersed polymer sample in water at a temperature of 20 °C, a viscosity of 1.002 cP, and a refractive index of 1.33 [51]. 

### 2.5. Fourier Transform Infrared Spectroscopic Analysis

The Fourier transform infrared spectroscopic (FTIR) spectra of the polymers particles were recorded with an FTIR spectrometer in the spectral range 4000–500 cm^−1^.

### 2.6. TGA Analysis

Thermal gravimetric analysis (TGA) investing the PGMA and SPGMA-based polymers was carried out using a Shimadzu Thermal Analyzer 50. (Kyoto 604-851, Japan).

### 2.7. Scanning Electron Microscopic Analysis

The scanning electron microscope (SEM) image of the polymer particles was obtained by placing the particles onto carbon tape-attached aluminum SEM stubs after coating them with gold to a few nanometer thicknesses under vacuum.

### 2.8. Preparation of Basic Methylen Blue Solution

Methylene blue (MB) stock solution was prepared by dissolving 0.1 g in 1000 mL distilled water using a magnetic stirrer. The MB concentration in the solutions after completing of the MB immobilization process was determined by measuring their absorbance in a 1cm light-path cell at a Max wavelength of 665 nm using UV- Visible spectrophotometer (T70+ PG Instruments).

### 2.9. Standard Curve of MB Concentration

Varied MB solution concentrations from 0.1 ppm to 5 ppm were prepared. The samples’ absorbance (A abs) was measured using a UV-Visible spectrophotometer and plotted against their concentrations. From the slope, we can derivative the Constant, equal to (1/Slope). The standard curve of MB concentrations is presented in Figure 1. The constant has been calculated from the curve’s slope and was found to be 4.65.

### 2.10. Methylene Blue-Polymers Composites Formation (Immobilization Process)

The methylene blue-polymers composites formation process was performed through immobilization experiments in a batch process using MB aqueous solution. The MB immobilization was performed by mixing 0.1 g of PGMA-based polymers with 10 mL of 10–40 ppm MB. The mixture was agitated at RT using a magnetic stirrer for 30 min, then centrifuged at 12,000 rpm to separate the matrix of the liquid phase. The MB concentration at ppm, before and after the immobilization, for each solution was determined by measuring the absorbance at the maximum wavelength (ʎ_*max*_ = 665 nm) using a UV-VIS spectrophotometer and multiplying by constant extracted from the slope of the standard curve. The MB-polymers composites composition as (mg/g) was calculated according to the following formula:MB-polymers composites composition (mg/g) = V (C_0_ − C_t_)/M(2)

C_0_ and C_t_ are the MB initial and final concentrations at definite immobilization time, Vis the volume of the MB solution (L), and Mis the mass of the PGMA and SPGMA-based polymers (g).

### 2.11. Chromium (VI) and Manganese (VII) Ions Removal

Synthetic dichromate (Cr VI) and permanganate (Mn VII) solutions, 20 mL, with varying concentrations, 2–8 ppm, were mixed with 0.1 g of MB-SPGMA composite at room temperature for 3 h, then separated by centrifugation under 12,000 rpm for 30 min, used in batch adsorption experiments. The Cr (VI) and Mn (VII) concentration at ppm, before and after the adsorption, for each solution was determined by measuring the absorbance at the maximum wavelength (ʎ*_max_* = 380 nm and 550 nm) using UV-VIS spectrophotometer and multiplied by constant extracted from the slope of the standard curve [52,53]. The adsorption capacity was calculated according to Equation (2). C_0_ and C_t_ are the metal ions’ initial and final concentrations at definite immobilization time, V is the volume of the metal ions solution (L), and M is the mass of the MB-SPGMA composite polymer (g).

## 3. Results and Discussion

### 3.1. Polymerization Process

The impact of the comonomers’ composition variation on the polymerization process has been studied. From the polymerization yield (%) results (Table 1), the polymerization yields are 89% and 100% for parent monomers; MMA and GMA. From the table, it is clear that the increase in the MMA percentage has an effect. The copolymers’ yield ranged between 86% and 93% depending on the MMA percentage in the feeding comonomers solutions.

### 3.2. Methylene Blue-Polymers Composites Formation (Immobilization Process)

The composition of both MB-PGMA-co-PMMA and MB-SPGMA-co-PMMA composites is tabulated in Table 2. First, it is clear from the inspection of the results that maximum immobilization capacity is assigned to the SPGMA polymers. Second, the variation of the PGMA content in the copolymers, sulfonated or not, is the determining factor affecting the immobilization capacity. The induced sulfonic ionic sites contributed mainly to the immobilization process through a chemo-sorption step for sulfonated samples. The PMMA moieties’ contribution to MB immobilization is mainly a physical one. Indeed, PMMA particles show enhancement in the immobilization capacity after the sulfonation step by three folds compared to the native one. That enhancement could be referred to some adsorbed sulfite groups remaining from sodium sulfite treatment. Based on the results mentioned above, the MB molecules immobilization process was found maximum using SPGMA polymer and further used in the following investigations.

Accordingly, the effect of variation of the MB concentration, polymer matrix dosage, immobilization time and temperature, and agitation speed on the composition of MB-SPGMA composite was monitored, and the results are presented in the following.

#### 3.2.1. Methylene Blue Concentration

Figure 2 shows the effect of variation of the MB concentration between 8 and 40 ppm on the MB-SPGMA-co-PMMA composites composition. The effect of the initial dye concentration factor depends on the immediate relation between the dye concentration and the available binding sites on a matrix surface [9]. Usually, the percentage of dye immobilization decreases with an increase in initial dye concentration, which may be due to the saturation of immobilization sites on the matrice surface [54]. As a result, there will be unoccupied active sites at low concentrations on the matrix surface. When the initial dye concentration increases, the active sites required to immobilize the dye molecules disappear [55]. In this work, the increase in the initial MB concentration will cause an increase in the MB content of the MB-SPGMA-co-PMMA composites, and this may be due to the high driving force for mass at a high initial dye concentration [56]. That indicates a high number of active sites relative to the number of MB molecules in the liquid phase of all the MB concentrations used where free active sites are still available. This postulation has been confirmed by the linear increase of the MB content of the MB-SPGMA-co-PMMA compositesto reach the highest value; 3.94 mg/g.

#### 3.2.2. SPGMA Polymer Dose

The SPGMA polymer amount increment has negatively affected the amount of immobilized MB amount s seen in Figure 3. The increment of the available active sites for the immobilization process explains the noticed behavior where the immobilization percentage increased from 73% to 98%. As a result, the immobilization capacity decreased from 14.7 mg/g to 3.94 mg/g. This behavior agrees with previous results obtained by the author [51].

#### 3.2.3. MB Immobilization Time

The variation of the MB immobilization time from 5 to 30 min slightly affects the MB immobilized amount from 3.65 to 3.94 mg/g (Figure 4). This behavior agreed with previously published data by the author using amidoximated polyacrylonitrile particles [57] and OPA-pyrazole-g-PGMA particles [29] for the immobilization of MB dye. In addition, a very fast equilibrium was achieved due to many available exchange sites relative to the MB molecules in the liquid phase.

#### 3.2.4. Agitation Speed

The variation of the agitation speed shows the same behavior, while no noticeable effect was observed with variation from 150 rpm to 300 rpm. That is an indication of the external mass diffusion absence due to the surface immobilization process nature, which is mainly controlled by the ratio between the number of active sites available on the polymer particles’ surface for the immobilization process of MB molecules from the solution (Table 3). This finding is contrary to other published results [29].

#### 3.2.5. MB Immobilization Temperature

Finally, Figure 5 shows the effect of varying the MB immobilization temperature on the MB content of the MB-SPGMA composites composition. The figure clearly shows that elevation of the immobilization temperature has a negative effect. The negative behavior upon elevation of the temperature indicates the exothermic nature of the MB immobilization process (Figure 5). This trend agrees with that obtained earlier using amidoximated crosslinked polyacrylonitrile particles [51]. The obtained results are an advantage since the dye immobilization process does not need additional heating or other costs. This behavior may be referred to as the acceleration effect of temperature on the dye molecules’ immobilization on the surface of SPGMA particles. This fast initial step reduces the concentration gradient between the MB dye liquid and polymer solid phases. MB concentration limitation from one side and high concentration of exchange sites over the surface of the particles on the other side contributes significantly to this behavior. The absence of a pore diffusion process also eliminates the effect of temperature [57].

### 3.3. Matrix Characterization

The effect of the monomers composition on the particle size of the developed copolymers has been monitored after the sulphonation step, Figure 6. The figure clearly shows that the PMMA pure polymer has the smallest particle size; 280.7 nm, while the SPGMA has a larger particle size, 430 nm. The incorporation of MMA with 25% in the monomer composition reduces the particle size of the formed SPGMA-co-PMMA to 148.0 nm. A further increase of the MMA monomer percentage to 50% sharply increases the formed SPGMA-co-PMMA to 755.2 nm, followed by a sharp decrease with a further increase of the MMA monomer percentage to 75%, reaching 180.0 nm. This behavior may be explained by the type of the formed copolymers and the steric effect of the introduced sulphonate groups. The obtained observations could be attributed mainly to the SPGMA content of the prepared copolymers. According to Toshio Kakurai [58], the epoxy content of the prepared PMMA-co-PGMA decreased directly with the GMA concentration in the comonomer composition. The maximum was obtained with 50:50 (*v*/*v*). These results support the changes in the developed SPGMA-co-PMMA particles size, taking into account the effect of introducing hydrophilic sulphonic groups into the epoxy pendants of PGMA content and, subsequently, the adherent water molecules, which directly causes swelling of the SPGMA particles. Accordingly, it was expected to have the largest particle size of the SPGMA. The partial sulphonation of the epoxy pendant rings of the SPGMA particles may be the logical explanation.

The effect of the monomers composition on the polymer’s particle morphology has been monitored, in Figure 7. The parent polymers, PMMA and SPGMA, have two different morphology structures. PMMA particles are smaller than the SPGMA, confirming the particle size results obtained previously in Figure 6. Both of the two polymers’ particles are irregular. PMMA particles are aggregated, while SPGMA particles have a flacked shape. Incorporating both monomers in different ratios affects the copolymers’ particle morphology. Increases the MMA monomer ratio and changes the flacks’ shape of the SPGMA particles to irregular particle shape. The only exceptional case appears in the copolymer particles produced from MMA:GMA equal volume ratio, 50%:50%, where the copolymer particles have the largest particle size, as confirmed by Figure 6.

The typical IR spectrum of pristine PMMA with characteristic peaks below 2000 cm^−1^ is shown in Figure 8A. The peak at 1720 cm^−1^ could be assigned to the C=O stretching of the esters. The absorption bands at 1271, 1240 cm^−1^, and 1189, 1141 cm^-1^ could be assigned to C–O, and C–O–C stretching, respectively. Because the PGMA polymer has a similar molecular structure to PMMA, most of the characteristic peaks discussed above also appear in the spectrum of PGMA-co-PMMA. It can be seen that a new and distinct absorption peak at 908 cm^−1^ appears in the spectrum of PGMA-co-PMMA (Figure 8B), which could be assigned to the epoxy group stretching. However, there is no noticeable absorption peak at the same wave number for the pristine PMMA [59]. As a result of the sulfonation process, the characteristic absorption band of the sulphonate group at 1050–1060 cm^−1^ was recognized for sulphonated samples. Since the epoxy groups in PGMA and PGMA-co-PMMA reacted with Na_2_SO_3_ to form PGMA-SO_3_Na, the epoxy groups would produce -OH groups. The figure shows that the intensity of the -OH peak (3500 cm^−1^) increases as the concentration of GMA increases in the feeding polymerization solution [49].

On the other hand, TGA thermo-grams (Figure 9) showed the weight loss of samples at temperature of 120 °C due to water evaporation. The variation of water loss due to the sulphonation process was observed for SPGMA and SPGMA-co-PMMAsamples. Remarkable thermal stability was observed for the sulphonated samples. A positive shift of the characteristic thermogram of PGMA starting at temperature of 240 °C to a higher temperature range 260–280 °C was recognized with an increase in the PGMA content of the sulfonated copolymers [50].

### 3.4. Dichromate and Permanganate Metal Ions Removal

Selected MB-SPGMA composite adsorbent with a composition 27.32 mg/g has been used for the first time in treating synthetic contaminated water with dichromate (Cr VI) and permanganate (Mn VII) ions under batch conditions. Synthetic contaminated water with various metal ions concentrations, 2–8 ppm, has been used in the study, Figure 10 and Figure 11. In general, the adsorption capacity of the MB-SPGMA composite adsorbent increased exponentially with an increase in the metal ions concentration. The capacity and rate of removal in the case of permanganate ions, Figure 11, is higher than the dichromate counter ions. Figure 10 shows that the adsorption capacity of MB-SPGMA composite adsorbent increased from 0.067 mg/g to 0.57 mg/g with an increase of the dichromate ions concentration from 2 ppm to 8 ppm. The curve shows two stages. The first stage is an almost linear increase of the adsorption capacity with an increase of the dichromate ions concentration up to 6 ppm. The second stage is an exponential increase of the adsorption capacity with an increase of the dichromate ions concentration from 6 ppm to 8 ppm, where the adsorption capacity increased by about two folds to reach 0.57 mg/g. Mohy Eldin et al. [52] found that the adsorption capacity of 0.2 g commercial Amberlite IRA 420 anion exchanger treating 50 mL Dichromate solution for 30 min in RT at pH 5.6 with a stirring rate 200 rpm reached 1.73 mg/g. The superiority of the commercial Amberlite IRA 420 anion exchanger may be referred to as the pH of the dichromate solution, which is lower than that used in this study; pH 7.4. On the other hand, Figure 11 shows that the adsorption capacity of MB-SPGMA composite adsorbent increased from 0.044 mg/g to 0.88 mg/g with an increase of the permanganate ions concentration from 2 ppm to 8 ppm. The curve shows two stages. The first stage is a linear increase of the adsorption capacity with an increase of the permanganate ions concentration up to 4 ppm. The second stage is an exponential increase of the adsorption capacity with an increase of the permanganate ions concentration from 4 ppm to 8 ppm, where the adsorption capacity increased by about ten folds to reach 0.88 mg/g. Mohy Eldin et al. [53] obtained the same adsorption capacity (0.93 mg/g) using 0.1 g of wet Amberlite IRA 420 with 10 mL of 100 ppm permanganate. The mixture was agitated at R·T using a magnetic stirrer for 30 min. The MB-SPGMA composite adsorbent shows superiority to the commercial Amberlite IRA 420 anion exchanger, which has the same adsorption capacity with treating 8 ppm permanganate solution compared with 100 ppm permanganate solution used in the case of using commercial Amberlite IRA 420 anion exchanger. The adsorption rate and capacity of the permanganate metal ions are higher than the dichromate metal ions. This superiority could be referred to as the metal ion radius difference, which directly affects the affinity of MB-SPGMA composite adsorbent towards the permanganate metal ions. The pH of the metal ions solution ranged between 7.0 and 8.0, which affects the charge of the MB-SPGMA composite adsorbent mainly by the deprotonation process of the nitrogen atom in the MB immobilized molecules, and finally, the hydrophilicity and porosity of the MB-SPGMA composite adsorbent.

## 4. Conclusions

Methylene blue-Polymers composites novel adsorbents have been developed by immobilizing MB azo dye onto both PGMA-co-PMMA and SPGMA-co-PMMA particles. The MB content of the MB-polymers composites has been monitored with the variation of the MB immobilization conditions. On the other hand, the sulfonation process of the copolymers’ PGMA content improves the MB content of the MB-polymers composites. Therefore, the PGMA content is determined to control the MB content of the MB-polymer composites. The MB content of the MB-polymers composites using SPGMA particles reached 15 mg/g. The immobilization process was very fast, with more than 90% of MB being immobilized within 5 min.

On the other hand, the agitation speed over 150 rpm was found to have of neglect able effect. Elevating the immobilization temperature from 25 °C to 60 °C reduced the MB content of the MB-SPGMA composites by about 22.5%. FTIR and TGA analyses monitored the effect of the comonomers’ composition on the chemical structure. In contrast, the physical changes were monitored by changes in the copolymers’ particle size and morphology. Finally, the capability of the developed MB-SPGMA composite in removing dichromate metal ions (Cr VI) and permanganate metal ions (Mn VII) has been proven for the first time in very mild adsorption conditions; room temperature and pH range 7.0–8.0. Further optimization of the adsorption conditions, especially the metal ions solution pH, concentration, temperature, agitation speed, and the adsorbent dose, should be investigated. The MB content of the MB-SPGMA composite is expected to be a determining factor.

In conclusion, the obtained data open a new area of research to benefit from the use of an adsorbent loaded with the first contaminant, such as MB in our study, in the removal of a second contaminant, such as dichromate and permanganate in our study, which reduces the cost of the wastewater treatment process of the industrial fluents.

## Figures and Tables

**Figure 1 polymers-14-04672-f001:**
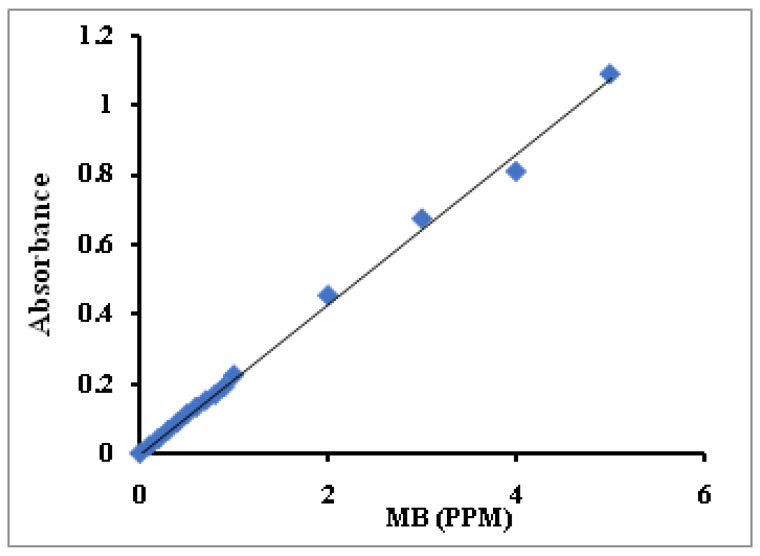
Standard curve for MB concentration.

**Figure 2 polymers-14-04672-f002:**
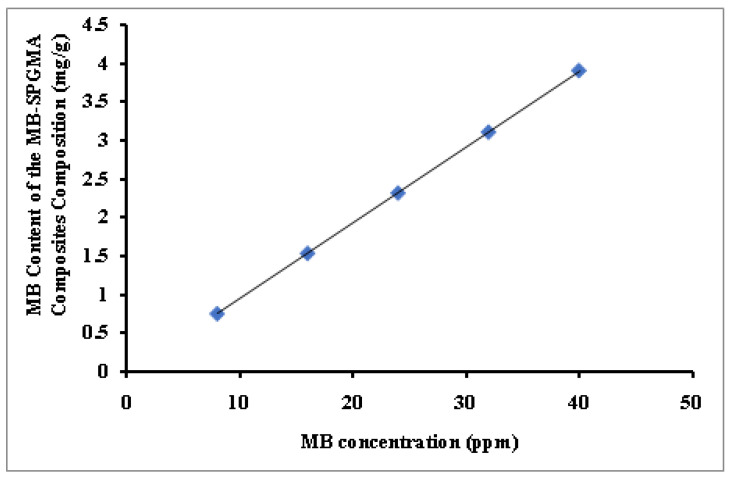
Effect of the MB concentration on the MB content of the MB-SPGMA composites composition.

**Figure 3 polymers-14-04672-f003:**
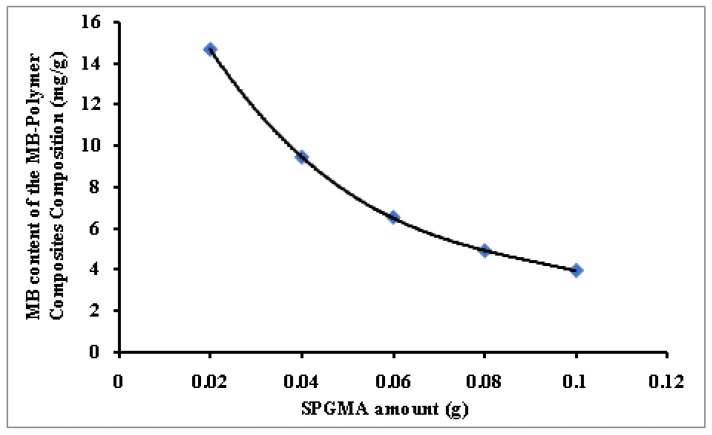
Effect of the SPGMA polymer dosage on the MB content of the MB-SPGMA composites composition.

**Figure 4 polymers-14-04672-f004:**
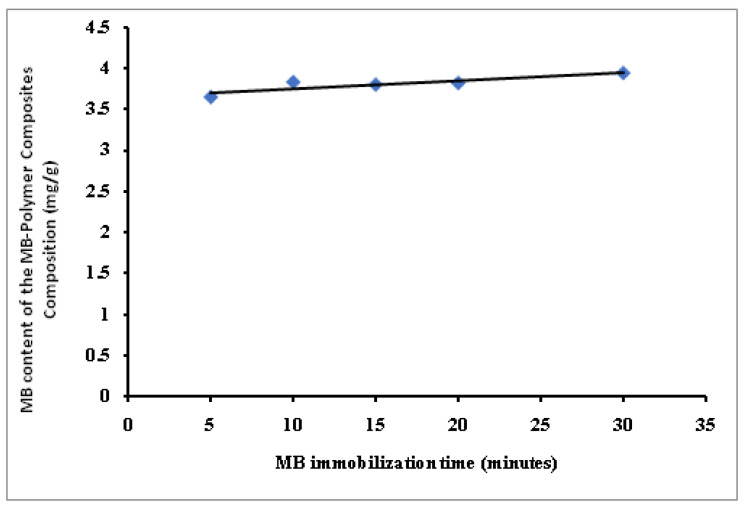
Effect of the MB immobilization time on the MB content of the MB-SPGMA composites composition.

**Figure 5 polymers-14-04672-f005:**
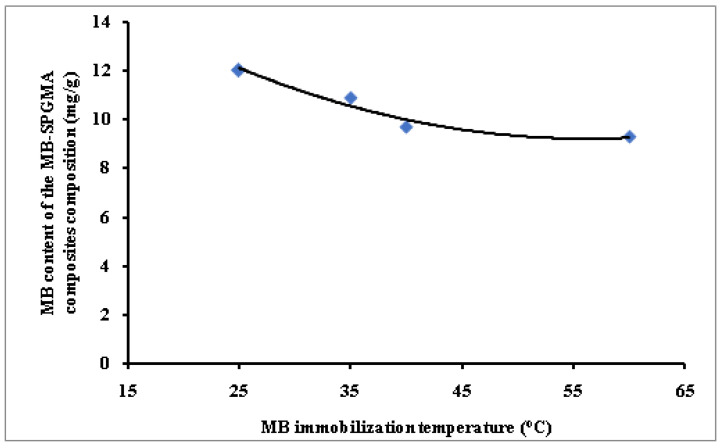
Effect of the immobilization temperature on the MB content of the MB-SPGMA composites composition.

**Figure 6 polymers-14-04672-f006:**
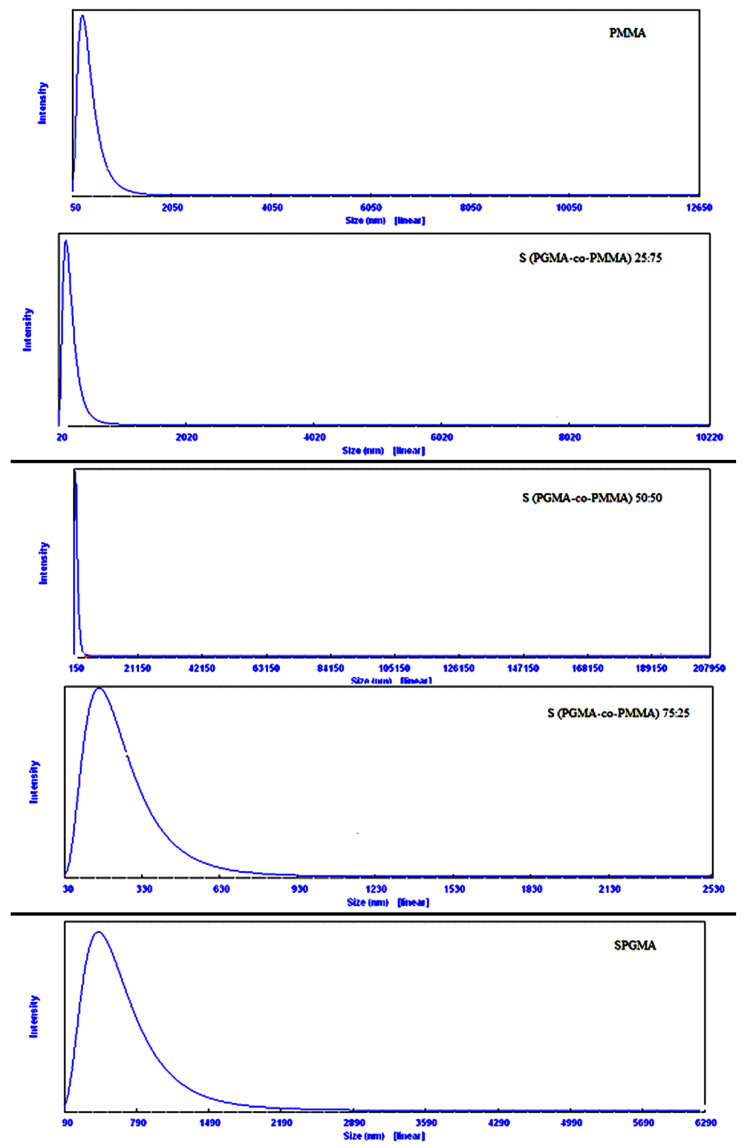
Effect of the comonomer composition on the particles size of the SPGMA based copolymers.

**Figure 7 polymers-14-04672-f007:**
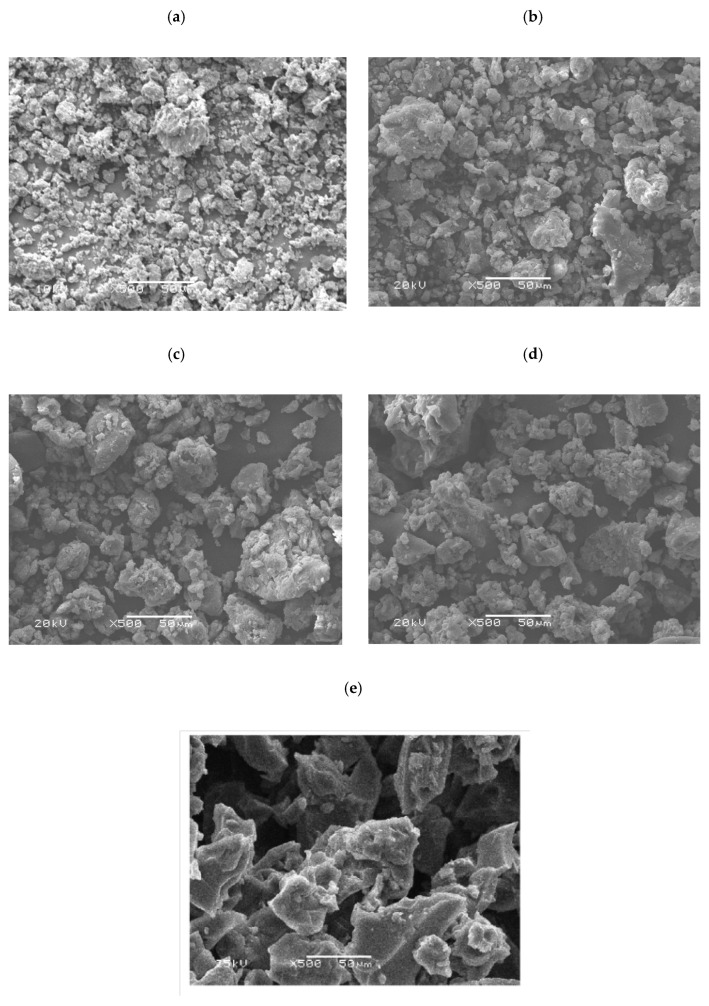
SEM photographs of: (**a**) PMMA, (**b**) SPGMA-co-PMMA; 25:75, (**c**) SPGMA-co-PMMA; 50:50, (**d**) SPGMA-co-PMMA; 75:25 copolymers, and (**e**) S PGMA.

**Figure 8 polymers-14-04672-f008:**
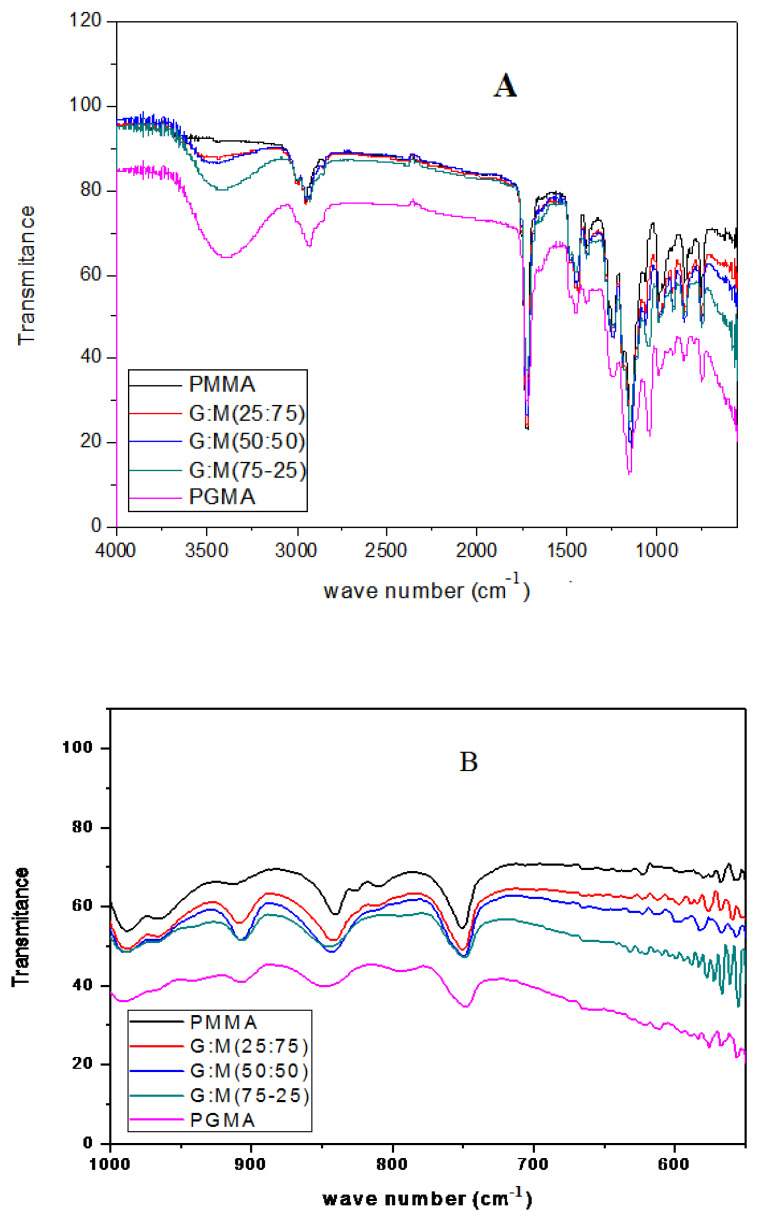
FT-IR spectra of PMMA, SPGMA-co-PMMA copolymers, and SPGMA; (**A**) 400-4000 cm^−1^, and (**B**) 400-1000 cm^−1^.

**Figure 9 polymers-14-04672-f009:**
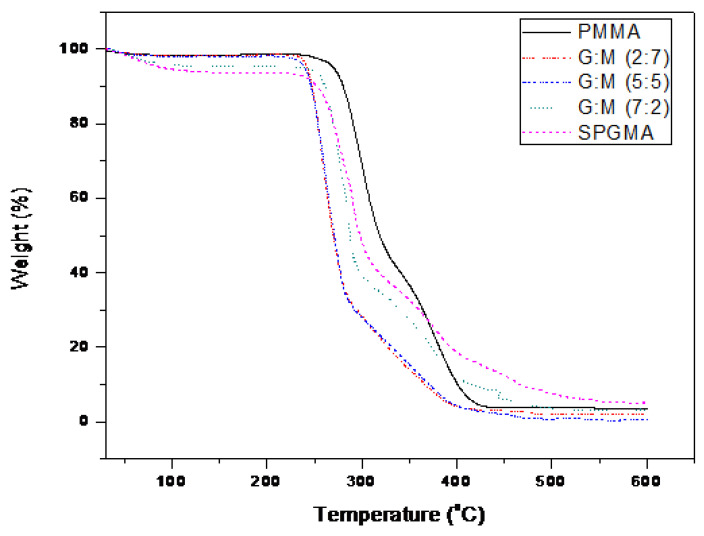
TGA thermo-gram of PMMA, SPGMA-co-PMMA copolymers, and SPGMA.

**Figure 10 polymers-14-04672-f010:**
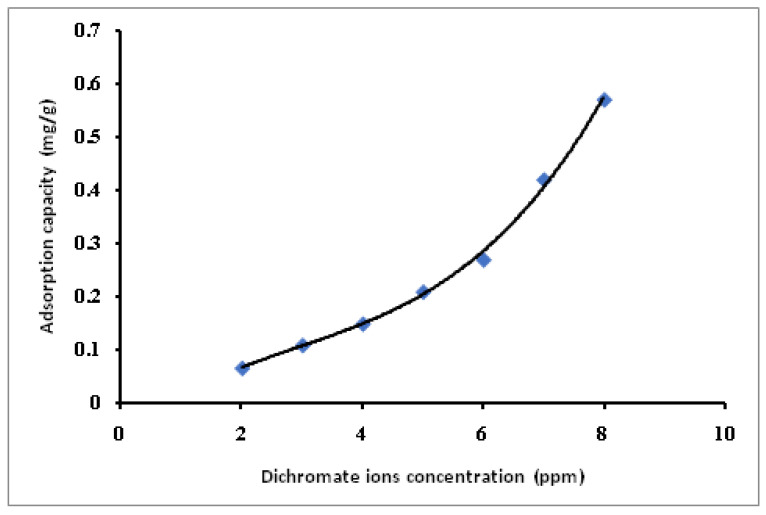
Effect of the dichromate metal ions concentration on MB-SPGMA composite adsorbent adsorption capacity.

**Figure 11 polymers-14-04672-f011:**
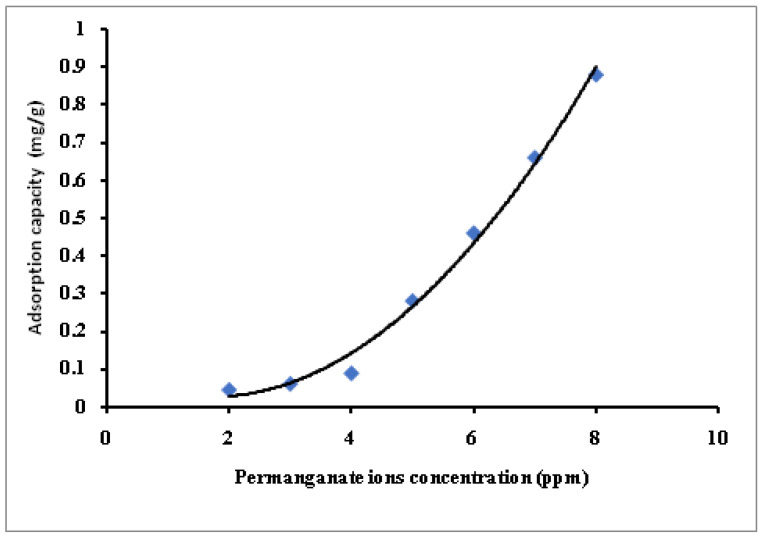
Effect of the permanganate metal ions concentration on the adsorption capacity of MB-SPGMA composite adsorbent.

**Table 1 polymers-14-04672-t001:** The polymerization yield (%).

GMA (%)	MMA (%)	Polymerization Yield (%)
100	0	100
0	100	89
75	25	85.6
50	50	92.3
25	75	93.5

**Table 2 polymers-14-04672-t002:** The composition of both MB-PGMA-co-PMMA and MB-SPGMA-co-PMMA composites.

Comonomers Composition (GMA:MMA) (*v*/*v*)	MB-PGMA-Co-PMMA (mg/g)	MB-SPGMA-Co-PMMA (mg/g)
100:0	0.9738	0.974
0:100	0.0935	0.258
75:25	0.255	0.978
50:50	0.4700	0.974
25:75	0.1676	0.853

**Table 3 polymers-14-04672-t003:** Effect of agitation speed on the MB content of the MB-SPGMA composites composition.

Agitation Speed (rpm)	150	200	250	300
MB content of the MB-SPGMA composites composition (mg/g)	3.91	3.88	3.89	3.88

## Data Availability

Not applicable.

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
