# Peer review of "Development of Azo Dye Immobilized Poly (Glycidyl Methacrylate-Co-Methyl Methacrylate) Polymers Composites as Novel Adsorbents for Water Treatment Applications: Methylene Blue-Polymers Composites"

_polymers, 2022, doi:10.3390/polym14214672_

Round 1
Reviewer 1 Report
- The paper is interesting. The paper deals with the introduction of (PGMA-co-PMMA), and (SPGMA-co-PMMA) adsorbents for water treatment applications. I have the following comments :
- The authors must enhance the writing of the paper, for example, threatens in line 38 to threat, criteria in line 52 to criteria s.
- Correct the way that references are cited, for example, [1.2] in line 37 and [4.21] to [4-21].
- The authors must explain further the physical reasons behind their observations about the changes in particle size.
Author Response
Comments and Suggestions for Authors
- The paper is interesting. The paper deals with the introduction of (PGMA-co-PMMA), and (SPGMA-co-PMMA) adsorbents for water treatment applications.
Answer: Thanks for your support.
I have the following comments:
- The authors must enhance the writing of the paper, for example, threatens in line 38 to threat, criteria in line 52 to criterias.
Answer: The entire manuscript has been revised and the needed corrections and modifications have been made.
- Correct the way that references are cited, for example, [1.2] in line 37 and [4.21] to [4-21].
Answer: The references have been revised according to the journal style.
- The authors must explain further the physical reasons behind their observations about the changes in particle size.
Answer: An interpretation of the changes in particle size has been given in lines 278-293 as following;
3.3. Matrix characterization
The effect of the monomers composition on the particle size of the developed copolymers has been monitored after the sulphonation step; Figure 6. The figure clearly shows that the PMMA pure polymer has the smallest particle size; 280.7 nm, while the SPGMA has a larger particle size; 430 nm. The incorporation of MMA with 25% in the monomer composition reduces the particle size of the formed SPGMA-co-PMMA to 148.0 nm. Further increase of the MMA monomer percentage to 50% sharply increases the formed SPGMA-co-PMMA to 755.2 nm, followed by a sharp decrease with a further increase of the MMA monomer percentage to 75%, reaching 180.0 nm. This behavior may be explained by the type of the formed copolymers and the steric effect of the introduced sulphonate groups. The obtained observations could be attributed mainly to the SPGMA content of the prepared copolymers. According to Toshio Kakurai [58], the epoxy content of the prepared PMMA-co-PGMA decreased directly with the GMA concentration in the comonomer composition. The maximum was obtained with 50:50 (V/V). These results support the changes in the developed SPGMA-co-PMMA particles size, taking into account the effect of introducing hydrophilic sulphonic groups into the epoxy pendants of PGMA content and, subsequently, the adherent water molecules, which directly causes swelling of the SPGMA particles. Accordingly, It was expected to have the largest particle size of the SPGMA. The partial sulphonation of the epoxy pendant rings of the SPGMA particles may be the logical explanation. [58] Kakurai, T.; Yoshida, E.; Noguch, T. The Reaction of Glycidylmethacrylate Copolymers with Amines. Kobunshi Kagaku 1968, 25, 413-18.
Reviewer 2 Report
The authors need to improve the English Language before submitting the manuscript for consideration of publication. TOO MANY language problems.
Author Response
Comments and Suggestions for Authors
The authors need to improve the English Language before submitting the manuscript for consideration of publication. TOO MANY language problems.
Answer: Thanks for the reviewer's comment and concern. The authors have revised the language of the entire manuscript and apologize for any inconvenience.
Reviewer 3 Report
1. The harmful effects of both dyes and metal ions must be added to the introductory part.
2. What is the need for MB immbolization?
3. Novelty of the work must be highlighted.
4. Adsorption models Langmuir and Freundlich should be investigated suggesting the usage of following citations
https://doi.org/10.1016/j.chemosphere.2021.133232.
Author Response
Comments and Suggestions for Authors
- The harmful effects of both dyes and metal ions must be added to the introductory part.
Answer: The harmful effects of both dyes and metal ions have been added to the introductory part with the proper references as following;
“Textile dyes are considered the most threatening source among other dyes [3-4]. The negative impact of releasing colors in the water system ranges from direct ones on aquatic life, and indirect ones on humankind's life. Methylene blue (MB) is a cationic dye commonly used worldwide as a coloring material [5]. Accordingly, corrective action to remove dyes from wastewater is needed. Different techniques have been used to remove dyes from wastewater, including physical, chemical, and even biological [6]. Among these techniques, physical and chemical ones have shown to be the most compromising techniques from many points of view [7-10]. The sustainability and cost-effective characteristics have driven the research for exploring alternative adsorbents for the last decades. Many publications have been published on the removal of different dyes [11-13]. Physiochemical technique attracts much attention among different techniques [14-21]. The physical adsorption of the dyes or chemicals bound over the surface of the adsorbent surface through electron exchange is the responsible interaction in the removal process [22]. Different factors affect the adsorption efficiency. Some of them are related to the structure of the adsorbent, and others are related to the operational conditions [23]. The adsorption technique is generally the most favored in the dye removal process based on many criteria [7-24]. In this context, activated carbon has shown a wide application [25]. However, different polymer-based adsorbents such as carboxylated alginate beads [26], grafted cotton fabrics [27], Chitosan Schiff Base Derivatives [28], and Phosphoric Acid Doped Pyrazole-g-Polyglycidyl Methacrylate [29] have been investigated. Equally, environmental pollution with heavy metals is pervasion worldwide with advances in the industry. Many heavy metals like nickel, copper, cadmium, manganese, and chromium are the most familiar toxic heavy metals used and the prevalence of environmental contaminants [30-31]. Low concentrations of those metals are essential as enzymes' co-factors, while high concentrations cause high toxicity to the living cells by inhibiting metabolism.
Potassium permanganate is commonly used in multidiscipline processes as a potent oxidizing agent for the oxidative treatment of many organic and inorganic compounds in soil and water solutions [32-33]. To our knowledge, few publications addressed the removal of permanganate ions. Adsorption is considered to be a cheap and efficient method for the removal of Mn (VII) from wastewater using different adsorbents such as activated orange peel powder [34], activated carbon [35-36], Prosopis cineraria leaf powder [37], and Millet husk [38]. Chromium can exist mainly as Cr (VI) or Cr (III) in the natural environment. Cr (III) species are less soluble and more stable compared to Cr (IV) species which are highly soluble and mobile in aqueous solutions [39]. Chromium (VI) also has higher mobility than chromium (III); therefore, it has a more significant potential to contaminate the groundwater. The high risk of chromium (VI) is associated with its high reactivity and potential carcinogenic properties [40]. Acute exposure to Cr (VI) causes nausea, diarrhea, liver and kidney damage, dermatitis, internal hemorrhage, and respiratory problems [5]. Inhalation may cause acute toxicity, irritation, and ulceration of the nasal septum and respiratory sensitization (asthma) [41]. Ingestion may affect kidney and liver functions. Skin contact may result in systemic poisoning damage, severe burns, and interference with the healing of cuts or scrapes. If not treated promptly, this may lead to ulceration and severe chronic allergic contact dermatitis. Eye exposure may cause permanent damage. Adsorption is also considered to be a cheap and efficient method for the removal of Cr (VI) from wastewater using different adsorbents such as charcoal [42], activated carbon from various sources [43-45], polyaniline and its composites [46], and Chitosan [47].
[3] Ayad, M. M.; Abo El-Nasr, A. Adsorption of cationic dye (methylene blue) from water using polyaniline nano-tubes base. J Phys Chem C 2010, 114, 14377–14383.
[4] Wong,Y. C.; Szeto,Y. S.; Cheung,W. H.; McKay, G. Equilibrium studies for acid dye adsorption onto chitosan. Langmuir 2003, 19, 7888–7894.
[5] Baybars, A. F.; Cengiz, Q.; Mustafa, K. Cationic dye (methylene blue) removal from aqueous solution by montmorillonite. Bull Korean Chem Soc. 2012, 33, 3184–3190.
[6] Yagub, M. T.; Sen,T. K.; Ang, H. M. Equilibrium, kinetics, and thermodynamics of methylene blue adsorption by pine tree leaves. Water, Air, & Soil Pollution 2012, 223, 5267-5282.
[7] Sen,T. K.; Afroze, S.; Ang, H. M. Equilibrium, kinetics and mechanism of removal of methylene blue from aqueous solution by adsorption onto pine cone biomass of Pinus radiate. Water, Air, & Soil Pollution 2011, 218, 499-515.
[8] Mohammad, M.; Maitra,;S.; Ahmad, N.; Bustam, A.; Sen, T. K. Metal ion removal from aqueous solution using physic seed hull. J Hazard Mater 2010, 179, 363-372.
[9] Abd EI-Latif, M. M.; Ibrahim, A. M.; EI-Kady, M. F. Adsorption equilibrium, kinetics and thermodynamics of methylene blue from aqueous solutions using biopolymer oak sawdust composite. J Am Sci 2010, 6, 267-283.
[10] Yao, Z.; Wang, L.; Qi, J. Biosorption of methylene blue from aqueous solution using a bioenergy forest waste: Xanthoceras sorbifolia seed coat. Clean (Weinh) 2009, 37, 642–648.
[11] Yagub, M. T.; Sen, T. K.; Afroze, S.; Ang, H. M. Dye and its removal from aqueous solution by adsorption: a review. Adv Colloid Interface Sci 2014, 209, 172-184.
[12] Salleh,M. A. M.; Mahmoud, D. K.; Karim,W. A. W. A.; Idris, A. Cationic and anionic dye adsorption by agricultural solid wastes: A comprehensive review. Desalination 2011, 280, 1–13.
[13] Srinivasan, A.; Viraraghavan,;T. Decolorization of dye wastewaters by biosorbents: a review. J Environ Manage 2010, 91, 1915-1929.
[14] Leszczyn´ ska, M.; Hubicki, Z. Application of weakly and strongly basic anion exchangers for the removal of brilliant yellow from aqueous solutions. Desalin Water Treat 2009, 2, 156–161.
[15] Purkait, M. K.; Maiti, A.; DasGupta, S. Removal of congo red using activated carbon and its regeneration. J Hazard Mater 2007,145, 287-295.
[16] Hernandez-Montoya, V.; Perez-Cruz, M. A.; Mendoza-Castillo, D. I.; Moreno-Virgen, M. R.; Bonilla-Petriciolet, A. Competitive adsorption of dyes and heavy metals on zeolitic structures. J Environ Manage 2013,116, 213-221.
[17] Errais, E.; Duplay, J.; Elhabiri, M.; Khodja, M.; Ocampo, R. Anionic RR120 dye adsorption onto raw clay: Surface properties and adsorption mechanism. Colloids Surf A Physicochem Eng Asp 2012, 403, 69–78.
[18] Ofomaja, A. E. Equilibrium sorption of methylene blue using mansonia wood sawdust as biosorbent. Desalin Water Treat 2009, 3, 1–10.
[19] Reddy, M. C.; Sivaramakrishna, L.; Reddy, A. V. The use of an agricultural waste material, Jujuba seeds for the removal of anionic dye (Congo red) from aqueous medium. J Hazard Mater 2012, 203-204, 118-127.
[20] Sarioglu, M.; Atay, U.A. Removal of Methylene blue by using biosolid. Global Nest J 2006, 8, 113–120.
[21] Deniz, F.; Karaman, S. Removal of Basic Red 46 dye from aqueous solution by pine tree leaves. Chem Eng J 2011, 170, 67–74.
[22] Dawood, S.; Sen, T. K. Removal of anionic dye Congo red from aqueous solution by raw pine and acid-treated pine cone powder as adsorbent: equilibrium, thermodynamic, kinetics, mechanism and process design. Water Res 2012, 46, 1933-1946.
[23] Auta, M.; Hameed, B. H. Coalesced chitosan activated carbon composite for batch and fixed-bed adsorption of cationic and anionic dyes. Colloids Surf B Biointerfaces 2013, 105, 199-206.
[24] Poinern, G. E. J.; Senanayake, G.; Shah, N.; Thi-Le, X. N.; Parkinson, G. M. Adsorption of the aurocyanide, View the MathML source complex on granular activated carbons derived from macadamia nut shells – A preliminary study. Miner Eng 2011, 24, 1694–1702.
[25] Dawood, S.; Sen, T. K.; Phan, C. Synthesis and characterization of novel-activated carbon from waste biomass pine cone and its application in the removal of congo red dye from aqueous solution by adsorption. Water, Air, & Soil Pollution 2014, 225, 1-16.
[26] Mohy Eldin, M. S.; Elkady, M. F.; Abdel Rahman, A. M.; Soliman, E. A.; Elzatahry, A. A.; Youssef, M. E.; Eweida, B.Y. Preparation and characterization of imino diacetic acid functionalized alginate beads for removal of contaminates from waste water: I. methylene blue cationic dye model. Desalin Water Treatment 2012, 40, 15-23.
[27] Mohy Eldin, M. S.; Gouda, M. H.; Abu-Saied, M. A.; El-Shazly, Y. M. S.; Farag, H. A. Development of Grafted Cotton Fabrics Ions Exchanger for Dye Removal Applications: Methylene Blue Model. Desalin water treatment 2016, 57, 22049-22060.
[28] El-Sayed, E. M.; Tamer, T. M.; Omer, A. M.; Mohy Eldin, M. S. Development of Novel Chitosan Schiff Base Derivatives for Cationic Dye Removal: Methyl Orange Model. Desalin water treatment 2016, 57, 22632-22645.
[29] Mohy Eldin, M. S.; Aly, K.; Khan, Z. A.; · Meky, A. E.; Saleh, T. S.; Elbogamy, A. S. Development of Novel Acid-Base Ions Exchanger for Basic Dye Removal: Phosphoric Acid Doped Pyrazole-g-Polyglycidyl Methacrylate. Desalin water treatment 2016, 57, 24047–24055.
[30] Z. Aksu, In: Y.-S. Wong, N.F.Y. Tam, Algae for wastewater treatment, Springer-Verlag and Landes Bioscience, Germany, Biosorption of heavy metals by microalgae in batch and continuous systems 1998, p. 37–53.
[31] Dönmez, G. ; Aksu, Z. The effect of copper (II) ions on growth and bioaccumulation properties of some yeasts. Proc. Biochem. 1999, 35, 135-142.
[32] Ahmaruzzaman, M. d. Adsorption of phenolic compounds on low-cost adsorbents: A review. Advances in Colloid and Interface Science 2008, 143, 48–67.
[33] Waldemer, R. H.; Tratnyek, P. G. Kinetics of contaminant degradation by permanganate. Environ Sci Technol 2006, 40, 1055–1061.
[34] Gupta, V.; Kumari, S.; Virvadiya, C. Adsorption Analysis of Mn(VII) from Aqueous Medium by Activated Orange Peels Powder. Int Res J Pure &Applied Chem 2015, 9, 1-8.
[35] Zhang , K.; Li , C.; He, J.; Liu, R. Adsorption of permanganate onto activated carbon particles. Hua Xi Yi Ke Da Xue Xue Bao 1997, 28, 344-346.
[36] Mahmoud, M. E.; Yakout, A. A.; Saad, S. R.; Osman, M. M. Removal of potassium permanganate from water by modified carbonaceous materials. Desalin Water Treat 2016, 57, 15559-15569.
[37] Virvadiya, C.; Kumari, S.; Choudhary, V.; Gupta, V. Combined bio- and chemosorption of Mn(VII) from aqueous solution by PROSOPIS CINERARIA leaf powder. Eur Chem Bull 2014, 3, 315-318.
[38] Chaudhary, M. Use of Millet Husk as a Biosorbent for the Removal of chromium and Manganese Ions from the Aqueous Solutions. Int J Chem, Environ and Pharm Res 2011, 2, 30-33.
[39] Wan Ngah, W.S.; Hanafiah, M.A.K.M. Removal of heavy metal ions from wastewater by chemically modified plant wastes as adsorbents: A review. Bioresource Technol 2007, 99, 3935–3948.
[40] Waranusantigula, P.; Pokethitiyook, P.; Kruatrachue, M.; Upatham, E.S. Kinetics of cbasic dye (methylene blue) biosorption by giant duckweed (Spirodela polyrrhiza). Environ Pollut 2003, 125, 385–392.
[41] Mohan, D.; Pittman, C.U. Activated carbons and low cost adsorbents for remediation of tri- and hexavalent chromium from water. J Hazard Mater B 2006, 137 , 762–811.
[42] Varga, M.; Takács, M.; Záray, G.; Varga, I. Comparative study of sorption kinetics and equilibrium of chromium (VI) on charcoals prepared from different low-cost materials. Microchem J 2013, 107, 25–30.
[43] Srinivasan, K. Evaluation of Rice husk carbon for the removal of trace inorganic form water. Thesis Submitted to I.I.T Madras (1986).
[44] Iqbal, M.; Saeed, A.; Zafars, S. I. Hybrid biosorbent: An innovative matrix to enhance the biosorption of Cd(II) from aqueous solution. J Hazard Mater 2007, 148, 47-55.
[45] Malathi, S.; Srinivasan K.; Gomathi, M. Studies on the removal of Cr (VI) from aqueous solution by activated carbon developed from Cottonseed activated with sulphuric acid. Int J Chem Tech Res 2015, 8, 795-802.
[46] Ansari, R. Application of Polyaniline and its Composites for adsorption/Recovery of Chromium (VI) from Aqueous Solutions. Acta Chim Slov 2006, 53, 88–94.
[47] Jassal, P. S.; Raut V. P.; Anand, N. Removal of Chromium (VI) ions from Aqueous solution onto Chitosan and Cross-linked Chitosan Beads. Proc Indian Natn Sci Acad 2010, 76, 1-6.
- What is the need for MB immbolization?
Answer: MB immobilization (adsorption) has two achieved goals. The first one is the removal of MB from MB-contaminated wastewater. The second goal is to develop novel composite adsorbents having affinity sites for the removal of Cr (VI) and Mn (VII) metal anions from dichromate and permanganate contaminated waters as second contaminates.
- The novelty of the work must be highlighted.
Answer: The novelty of the work has been highlighted at the end of the introduction section as follows;
“The present study's novelty is the first emphasis on using one adsorbent to remove multi-contaminants from wastewater. This goal has been achieved as follows. Native PGMA and SPGMA-based copolymers have been immobilized with azo dye through the removal of Methylene Blue (MB) molecules as the first contaminates to develop novel composites adsorbents having an affinity for removal of Cr (VI) and Mn (VII) metal anions from dichromate and permanganate contaminated waters as second contaminates.”
- Adsorption models Langmuir and Freundlich should be investigated suggesting the usage of the following citations
https://doi.org/10.1016/j.chemosphere.2021.133232.
Answer: The authors thank the reviewer for his/her suggestion and agreed with it. A separate manuscript dealing with the isotherm, kinetic, thermodynamic, and simulation of the MB adsorption process is now under review by another journal.
Reviewer 4 Report
Before, I consider the acceptability of the present manuscript, an extensive editing (meaning, phrasing etc.,) of the English language, plus proper templating (references are not templated for MDPI, mL and not ml, multiply is not letter x, etc., etc.,) is mandatory.
Author Response
Comments and Suggestions for Authors
Before I consider the acceptability of the present manuscript, extensive editing (meaning, phrasing, etc.,) of the English language, plus proper templating (references are not templated for MDPI, mL and not ml, multiply is not letter x, etc., etc.,) is mandatory.
Answer: The authors thank the reviewer for his/her concerns about the language of the manuscript and the proper templating. The entire manuscript has been revised and the needed corrections and modifications have been made.
Round 2
Reviewer 2 Report
the authors have addressed issues mentioned. The reviewer suggest to publish it on polymers.
Author Response
Dear Reviewer
The authors thank you very much for your efforts in improving the manuscript.
Reviewer 3 Report
The followings points must be considered
1. Self-citations are not a good thing, I advise removing most of the self-citations and adding other citations.
2. Add this https://doi.org/10.1021/acs.langmuir.2c00702 in the introduction part.
3. The reference's formatting must be done to make them uniform.
Author Response
Dear Reviewer
Thank you very much for you valuable comments.
Please find below the answers to your comments.
Greetings,
Comments and Suggestions for Authors
The followings points must be considered
- Self-citations are not a good thing, I advise removing most of the self-citations and adding other citations.
Answer: Thank you very for the valuable comment which we are totally agreed with in case of inappropriate self-citation. Based on the reviewer comment, we make sure of using only the relevant references to the manuscript scientific content especially the self-citation.
Accordingly,
- The reference 28 has been omitted.
- Other self-citation references such as 27, 28, 29, 81, and 57 are related to the Methylene Blue (MB) removal by adsorption. While the references 49 and 50 are related to the polymerization and sulfonation processes of GMA. Finally, the references 52 and 53 are related to the adsorption removal of Cr (VI) and Mn (VII) metal anions. The total self-citation references are 9 per 59 total numbers of references.
- Add this https://doi.org/10.1021/acs.langmuir.2c00702 in the introduction part.
Answer: Thanks for the reviewer suggestion. The reference has been added as new [26]; [26] Khan, S. A.; Abbasi, N.; Hussain, D.; Khan, T. A. Sustainable Mitigation of Paracetamol with a Novel Dual-Functionalized Pullulan/Kaolin Hydrogel Nanocomposite from Simulated Wastewater. Langmuir 2022, 38, 8280-8295.
- The reference's formatting must be done to make them uniform.
Answer: Thanks for the reviewer comment. The references have been revised and uniformed according to the journal style.
Reviewer 4 Report
In this manuscript entitled “Development of Azo Dye Immobilized Poly(Glycidyl methacrylate-co-Methyl Methacrylate) Polymers Composites as Novel Adsorbents for Water Treatment Applications: Methylene Blue-Polymers Composites”, El-Aassar, Lohy-Eldin and co-workers have reported the synthesis of PGMA-co-PMMA, and SPGMA-co-PMMA polymers composites loaded with MB. The chemical structure and morphology of the developed composites was then studied through particle size, FTIR, TGA, and SEM analyses, and removal of Cr (VI), and Mn (VII) ion contaminants into water studied. Overall, the entire manuscript has been revised, and corrections as regard to the scientific soundness, and quality of presentation of the investigation performed. I thus recommend publication in Polymers, after the following minor corrections have been performed.
L27 (…were tested for …??)
L88-89: superscript 2+ and 3+ of Zn2+, Mn2+, and Cr3+
2.1 Materials: The source from where Cr ions have been purchased is not documented.
L124: at a temperature of …
Modify all temperatures with good template = XX + space + °C.
L145: sentence meaning??
Homogenize RT (RT or R.T.)
L228: space after of
Superscript -1 of cm-1.
L306: Subscript 2 and 3 of Na2SO3 and SO3Na
L346: space after 0.2
L354: space after 4 of 4 ppm
Reference section: Please use the good template, see https://www.mdpi.com/authors/references
Author Response
Dear Reviewer
Thank you very much for your valuable comments.
Please, find below the answers to the comments.
Greetings,
Comments and Suggestions for Authors
In this manuscript entitled “Development of Azo Dye Immobilized Poly(Glycidyl methacrylate-co-Methyl Methacrylate) Polymers Composites as Novel Adsorbents for Water Treatment Applications: Methylene Blue-Polymers Composites”, El-Aassar, Lohy-Eldin, and co-workers have reported the synthesis of PGMA-co-PMMA, and SPGMA-co-PMMA polymers composites loaded with MB. The chemical structure and morphology of the developed composites were then studied through particle size, FTIR, TGA, and SEM analyses, and the removal of Cr (VI), and Mn (VII) ion contaminants into the water studied. Overall, the entire manuscript has been revised, and corrections as regards the scientific soundness, and quality of presentation of the investigation performed.
I thus recommend publication in Polymers, after the following minor corrections have been performed.
L27 (…were tested for …??)
Answer: a correction has been made.
L88-89: superscript 2+ and 3+ of Zn2+, Mn2+, and Cr3+
Answer: a correction has been made.
2.1 Materials: The source from where Cr ions have been purchased is not documented.
Answer: The sources from where Cr and Mn ions have been cited are as follows;
Potassium dichromate (K2Cr2O7), minimum assay 99%, was supplied by Sigma Aldrich, Germany. Potassium permanganate (KMnO4), a minimum assay of 99%, was supplied by Sigma Aldrich, Germany.
L124: at a temperature of …
Modify all temperatures with good template = XX + space + °C.
Answer: All temperatures have been modified.
L145: sentence meaning??
Answer: The sentence has been reformate as follows: “The MB concentration in the solutions after completing the MB immobilization process was determined by measuring their absorbance in a 1 cm light-path cell at a Max wavelength of 665 nm using UV- Visible spectrophotometer (T70+ PG Instruments).”
Homogenize RT (RT or R.T.)
Answer: Homogenization has been made as RT.
L228: space after of
Superscript -1 of cm-1.
Answer: done.
L306: Subscript 2 and 3 of Na2SO3 and SO3Na
Answer: done.
L346: space after 0.2
Answer: done.
L354: space after 4 of 4 ppm
Answer: done.
Reference section: Please use the good template, see https://www.mdpi.com/authors/references.
Answer: Thanks for the reviewer's comment. The references have been revised and uniformed according to the journal style.